# The Surgical Benefit of Hysterolaparoscopy in Endometriosis-Related Infertility: A Single Centre Retrospective Study with a Minimum 2-Year Follow-Up

**DOI:** 10.3390/jcm9020507

**Published:** 2020-02-13

**Authors:** Atombosoba Adokiye Ekine, István Fülöp, István Tekse, Árpád Rúcz, Sara Jeges, Ágnes Koppán, Miklós Koppán

**Affiliations:** 1Department of Obstetrics & Gynaecology, Robert Károly Private Hospital, 1135 Budapest, Hungary; fulopdoc@gmail.com (I.F.); drtekse@gmail.com (I.T.); rucz1@hotmail.com (Á.R.); 2Faculty of Health Sciences, University of Pécs, 48-as tér 1, 7622 Pécs, Hungary; jeges@etk.pte.hu (S.J.); akoppan@laparo.hu (Á.K.); 3Department of Obstetrics & Gynaecology, University of Pécs, 48-as tér 1, 7622 Pécs, Hungary; koppan.miklos@laparo.hu

**Keywords:** endometriosis, infertility, hysterolaparoscopy surgery, pregnancy, quality of life

## Abstract

Aim: This study examined the fertility performance of women after combined hysterolaparoscopic surgical management of endometriosis. *Design:* This study is a hospital-based retrospective review. Materials and Methods: Data collected from the records of all patients presented with endometriosis-related infertility using a checklist designed for the purpose. *Result:* A total of 81.3% (370/455) of women who have had the desire to have children became pregnant during the study period after the surgery. Of those who became pregnant, all three-hundred-forty-seven patients were followed to the end of their pregnancies. A successful live birth occurred in 94.2% (327/347) of individuals, and pregnancy loss occurred in 5.8% (20/347). The mean patient age was 34.1 ± 4.1 years, and the average duration of infertility was 3.4 ± 3.3 years. Pregnancy occurred spontaneously in 39.5% (146/370) of patients, after artificial insemination (AIH) in 3.8% (14/370) of women, and after in vitro fertilization-embryo transfer (IVF-ET) in 56.8% (210/370) of cases. Patients aged ≤ 35 years had a higher chance of conception post-surgery—84% versus 77%, respectively (*p* = 0.039). Based on the modes of pregnancy, the timely introduction of an assisted reproductive technique (ART) demonstrated a significant effect on fertility performance postsurgery. Comparatively, this effect was 91.3% vs. 74.1% among the ≤35- and >35-year-old age groups, respectively. There was no significant difference in reproductive performance based on stages of endometriosis, nor in the other parameters evaluated. *Conclusion:* Our data are consistent with previous clinical studies regarding the management options of endometriosis-related infertility. Overall, the combined hysterolaparoscopy treatment is a very effective and reliable procedure, and is even more effective when combined with ART. It enhances women’s wellbeing and quality of life, and significantly improves reproductive performance.

## 1. Introduction

Infertility causes an immense psycho-emotional imbalance in many couples. Despite the enormous improvement in reproductive medicine in the past two or more decades, the challenges are still remarkable. Current indices have shown that childbearing age has increased, mostly in developed countries, in the recent past. Some of the causes of the delay in childbearing age include higher educational pursuits, industrialization, value change, and the increased participation of women in professional fields. Sadly, women in their prime childbearing age (between 25 and 40 years) are the most affected by endometriosis [1]. Moreover, despite extensive research, a direct correlation between endometriosis and infertility or subfertility has not been fully established. However, the possible association between the disease and infertility is becoming more apparent. Scientifically, fecundity is about 0.15 and 0.20 per month in average couples, whereas women with endometriosis have a reported range of 0.02–0.1 per month [2]. Other studies report lower live births in women with endometriosis compared to those without the disease [1,2]. Endometriosis is a common gynaecological disease, affecting over 70 million women worldwide, although the actual prevalence is unknown as the diagnosis of the disease is delayed in most cases [3]. However, some authors estimate a prevalence of between 2% and 10% for women in the general population and a staggering 20–50% of the female population with fertility issues [4,5,6]. It is common among young women of reproductive age, often associated with distressing and life-threatening conditions [7,8]. It is characterized or defined as the presence of endometrial-like tissues (glands or stroma) outside the endometrium of the uterus [6,9,10]. These ectopic endometrial-like tissues are morphologically similar to the normal endometrium. Consequently, they react to the ovarian hormonal influence, a process that induces a recurrent chronic inflammatory reaction [6,9,11]. Apart from its effect on fertility, endometriosis is associated with dysmenorrhea, dyspareunia and lower abdominal pain, and it sometimes causes dysuria and dyschezia, depending on the degree and localization [7,12]. Although the real pathogenesis of endometriosis remains unclear, there are numerous hypotheses [13]. These include altered immunity, coelomic metaplasia and metaplastic spread, retrograde menstruation and implantation. Recently, genetic predisposition of the disease has received attention [11,13,14,15,16]. Pathophysiological and biochemical investigations have shown that women with endometriosis exhibit an increased amount of peritoneal fluid, which is associated with high concentrations of prostaglandins, proteases and cytokines. These cytokines include inflammatory molecules, such as interleukin 1 (IL1), IL-6 and tumour necrosis factor (TNF), and angiogenic species like IL-8 and vascular endothelial growth factor (VEGF) [17,18,19]. These alterations in the pelvic floor may exert toxic effects on oocytes, sperm and embryos and may also impair tubal motility, notwithstanding the stage induced by endometriosis [20,21]. In the most advanced cases, in addition to the latter effect, distortion of the pelvic organs, caused by adhesions, can impair oocyte release, reduce tubal transportation by displacement or ultimate blockage and affect sperm migration [22].

Adenomyosis, another form of endometriosis, can cause pregnancy loss by interfering with implantations [23]. This mysterious disease can be detected—or suspected—from a patient’s history, serum markers, ultrasounds and other radiological imaging. However, the gold standard for diagnosis remains direct visualization by laparoscopy or laparotomy [24]. Infertility causes a loss of self-esteem in women, while endometriosis-associated infertility constitutes a significant burden on the quality of life of women and their families, as well as the healthcare system [7,25]. There were also other similar studies where hysteroscopy and laparoscopy were combined in patients with endometriosis-related infertility, with varying degree of success [1,4,26,27].

## 2. Methods

Ethical approval was obtained from the hospital ethical committee to access the patient database. This retrospective, single-hospital centre study was conducted at the Endoscopic and IVF unit of the Robert Károly Private Hospital, Budapest, from June 2010 to June 2018, using the hospital’s computerized database. This study aimed to determine the post-surgical performance of patients with infertility issues, with or without other endometriosis-related symptoms. The study sample comprised 533 women who attended the general gynaecological consulting unit and the infertility unit with infertility and other symptoms associated with endometriosis. A decision for surgical management was based on the symptoms and signs suggestive of endometriosis, such as patients’ complaints, previous medical history, ultrasonography findings and results from other investigative procedures, including magnetic resonance imaging (MRI), computed tomography (CT), serum markers. All of the patients were referred to the minimal access gynaecology (endoscopic) department of the hospital. All of the patients underwent simultaneously combined hysterolaparoscopy surgical management between January 2010 and December 2016. They were all diagnosed with different stages of endometriosis. Post-surgical follow-up lasted for a maximum of 2 years according to the date of surgery. Endometriosis stages were scored according to the revised American Fertility Society (rAFS) score. During the postoperative follow-up, 29 patients opted out of their intention to bear children for personal reasons, and the number lost to follow-up was 49. Finally, only 455 patients were able to complete the study. Before the surgery, all the patients had several infertility treatments, both artificial inseminations (AI) and artificial reproductive techniques in vitro fertilization (IVF) treatments at least for once. Immediately after surgery, some of the patients were advised to consider and proceed with the normal process of conception, while those with proven complete tubal blockage, or those who failed to become pregnant after 12 months, were referred to an in vitro fertilization (IVF) programme based on the surgical results. All of the patients were followed for a maximum of 24 months, with information being gathered either by the computerized hospital database or by direct electronic communication (email, telephone). The variables considered for this study included age, parity, type of infertility, duration of infertility, stage of endometriosis, mode of pregnancy, pregnancy outcome and other presenting symptoms (dysmenorrhea, dyspareunia, chronic pelvic pain, urinary symptoms). Ultrasonography findings, such as the presence of endometrioma, adenomyosis, tenderness, adnexal masses and mobility of the uterus, were also considered. All patients with primary and secondary infertility issues—who currently intended to have children—and patients who had operative or diagnostic hysterolaparoscopy and chromo-perturbation test and were diagnosed to have endometriosis (various degree or stages) were considered for the study. Women with pelvic inflammatory disease (PID), polycystic ovary syndrome (PCOS), adhesions from previous surgical interventions and infections were all excluded. The stage or severity of endometriosis was based on the surgical findings using the principle of the rAFS score, include powder-burn black, red, white, yellow, dark blue, brown and nonpigmented lesions. The size, depth and location of these lesions were also considered, the disease categorized into four stages [19].

All data entered into SPSS version 24. The collected data expressed as mean with standard deviation, frequency, percentage or cumulative percentage. Association between age, stage, mode of pregnancy, laparoscopic surgical management and fertility performance assessed using the chi-square test for categorical data or the two-sample *t*-test for continuous variables. We adopted a multiple binary logistical regression model to determine the statistical relationship (using age as control). A *p* value < 0.050 was considered to indicate a statistically significant difference.

## 3. Results

Out of the 504 patients with infertility issues prior the surgical management, 49 were lost to follow-up during the first 12–24-month median follow-up period, and thus were excluded from the study. Finally, 455 patients with a desire and interest in having children were evaluated in the final analysis. The patient age range was 25–46 years, with a mean of 34.3 ± 4.1 years. The overall cumulative pregnancy rate in this study was 81.3% (370/455). Figure 1 is illustrative as it shows how many people were in each age group, although it does not highlight the differences in success between age groups. However, it shows a slightly significant difference among the groups. Figure 1 and Figure 2 illustrate the relationship between pregnancy outcomes after endometriosis surgery by age group and by stages of endometriosis. While Figure 2 demonstrates the postsurgical fertility performance by the stages of the endometriosis but does not highlight differences or any significant outcome in success between the stages of endometriosis. There is no significant difference in the success rate of the four groups. Table 1, Table 2 and Table 3 analyze the frequency, and the number of patients, respectively, who became pregnant and those unable to achieve pregnancy after surgery. Table 1 illustrates the demographics, menstrual pattern, and stages of endometriosis in this study. Table 2, demonstrates some of the factors which may affect women’s wellbeing and quality of life; independent of the endometriosis-related painful distress, infertility that is associated with endometriosis and bleeding disorders, resolved with the aid of a combined hysteroscopy and laparoscopy surgery. Table 3 and Table 4 illustrate the effectiveness of the surgical procedure shown, by the fact that those who received one or more ART treatments before surgery became pregnant spontaneously—74.1% of the younger group, 73.6% of the elderly group (no significant difference), thus regardless of age. Those who received ART after surgery included 91.3% of the younger group and 83.0% of the older group (no significant difference between both groups). Table 5 is an illustration of the effect of different surgical procedures on fertility performance after a thorough evaluation of need and stage of endometriosis. A multiple binary logistical regression model was used to ascertain the relationship between whether pregnancy occurred and the investigated variables by controlling for age (≤35 versus >35 years). Patients who underwent the assisted reproductive technique (ART) after surgery had a better chance of pregnancy compared to those without post-surgery ART (odds ratio [OR] = 2.2; 95% confidence interval [CI] = 1.2–3.6). We analyzed the success rate of the surgery in the different (but comparatively homogenous) age groups with regard to fertility performances by considering post-surgical ART treatment. For patients who had several unsuccessful pre-surgical ART treatments, 74.1% (40/54) of women ≤ 35 years and 73.6% (39/53) of women > 35 years became pregnant without ART treatment post-surgery (*p* = 0.954). This finding shows that the fertility performance was not significantly influenced by the age of the women, but rather as a result of the total removal of visible lesions. However, when ART was included post-surgically and irrespective of the state of the patient—including those with other abnormalities (male infertility, etc.), delayed conception or complete tubal blockage—there were no significant differences in the pregnancy rate between the age groups: 91.3% in the ≤35-years-old group and 83.0% in the >35-year-old group. There was a significant difference in the mode of pregnancy outcome in the ≤35-year-old group when we compared those who became pregnant spontaneously to those by ART-assisted pregnancy—74.1% (40/54) versus 91.3% (73/80) post-surgery (*p* = 0.007; OR = 3.7; 95% CI = 1.4–9.8). Meanwhile, in >35-year-old group, there was no statistically significant difference in the mode of pregnancy outcome—83.0% (73/88) versus 73.6% (39/53) (*p* = 0.111)—at the end of the follow-up period.

## 4. Discussion

The ultimate goal of any medical intervention is to address the issues that affect the quality of life and socio-cultural well-being of the patient. Generally speaking, infertility is not a life-threatening medical condition.

However, some of the consequential effects could be life-threatening. Infertility affects women all over the world, and the socio-cultural stigma that surrounds it varies and can often result in family breakdown [28,29,30]. Many studies have shown an association between endometriosis and infertility, and this not overemphasized. Endometriosis was first discovered microscopically by Karl von Rokitansky in 1860; however, records have shown that it has been documented over 4000 years [8,31]. This retrospective study is consistent with other previous studies. The combined hysteroscopy–laparoscopy surgical treatment of endometriosis-related infertility improves fertility performance and restoration; therefore, it is gradually being considered to be among the best options currently available, irrespective of the few shortcomings associated with the procedure. Furthermore, a well-timed ART introduction also proved to further improve the reproductive performance of the affected women [32,33,34]. Our results demonstrate the significant benefits of the hysterolaparoscopic management of endometriosis-related infertility. The pregnancy rate improved considerably after the surgery, namely 81.3% (370/455), with a live birth rate of 94.2% (327/347). We also observed that the endometriosis stage did not significantly influence fertility performance in those women. These data also indicate that fertility performance is more dependent on surgical expertise and patient’s age rather than endometriosis stage. Consistent with these results, another review conducted by Jacobson et al. also concluded that the laparoscopic treatment of minimal and mild endometriosis improves pregnancy and live birth rates [32,33]. Meanwhile, Fuchs et al. recorded a high pregnancy rate (65%) within an 8.5-month postoperative follow-up period, of which 86.9% pregnancies resulted in deliveries [34,35]. For minimal and mild subfertility-related endometriosis, clinicians remain of the opinion that laparoscopic surgery treatment may increase the chances of future pregnancy and live birth [4,36,37]. Another study presented a pregnancy rate of 81.6% and 43.7% live births in stage I and II endometriosis, respectively, while a 56.7% pregnancy rate and 40.3% live birth rate was recorded in those with stage III and IV endometriosis, respectively, after 1–4 IVF/intracytoplasmic sperm injection (ICSI) treatments cycles [28,38].

However, with a relatively smaller study population, Słabuszewska-Jóźwiak et al. also reported that 20.75% of the patients became pregnant spontaneously, ending a live birth, without ART argumentation in the first 6 months of follow-up [39].

Nardo et al. reported a cumulative pregnancy rate of 23.2% after laparoscopic treatment with the Helica Thermal Coagulator for minimal and mild endometriosis [40]. Both study outcomes were similar to our report, as we recorded an overall 28.8% spontaneous pregnancy rate during the follow-up period. However, our study was broader because all endometriosis stages were involved [36,40,41,42].

A systematic review also emphasized the beneficial effect of laparoscopic surgery for the treatment of subfertility related to minimal and mild endometriosis [32,43,44,45]. Generally, the overall pregnancy outcome—irrespective of the stages of the endometriosis—improves after laparoscopic surgical intervention on endometriosis-related infertility [10,17,32,33]. The American Society for Reproductive Medicine (ASRM)’s classification of endometriosis has been reported, by many authors as a useful tool in predicting IVF success and fertility performance [43]. Additionally, the European Society for Human Reproduction and Embryology (ESHRE) recommends that lesions classified as moderately severe to severe have a better chance with IVF as the first line of treatment [6,39,44]. However, there was no significant difference among the endometriosis stages regarding fertility performance—in patients with expectant management or with those who opted for IVF—even without any anatomical deficiencies. Among our study population, 12% (61) were diagnosed as stage 1, 26.2% (132) as stage 2, 34.9% (161) as stage 3 and 19.8% (100) were stage 4. Explanations of the relatively improved fertility performance of this study may emanate from the combined hysteroscopy laparoscopy surgical and laser approach. This technique includes the use of a CO_2_ laser on dense peritoneal involvements for further surgical improvement. Furthermore, surgical expertise, which may not be the same in some of the previous studies, is another crucial factor, in addition to other co-joining factors, such as population, environment, personal factors, the efficiency of IVF centre etc., and individuals in those study groups. The combination of hysteroscopy and laparoscopic surgical manoeuvres in this study was necessary because of uterine abnormalities and other most largely tubal abnormalities which contribute to approximately 30–35% of female infertility [36,45]. Hysteroscopy is an efficient tool in the investigation and correction of intrauterine abnormalities, intratubal adhesions and hydrosalpinx, among other issues. The direct vision of the tubal os, chromo-tubation and peritubal adhesiolysis via the laparoscopy enhances and, consequently, promotes better chances of pregnancy compared to the painful hysterosalpingography (HSG) or hysterosalpingo contrast sonography (HYSCOY), both of which can provide false-negative results [36,39,46]. However, surgical and other management plans are required at different times due to the symptoms and manifestation complexity of endometriosis, with the differences in age, localization, desire for pregnancy and disease severity. In this centre, we deployed different surgical approaches, which ranged from electro-cauterisation of visible endometriosis implants, the excision/stripping surgical technique of the ovarian endometrioma and laser coagulation. The various surgical methods were employed to enhance the postoperative outcome and reduce the recurrence ratio. Other additional surgical manoeuvre included adhesiolysis and adenomyomectomy, in cases of Adenomyosis, similar to other studies [22,39,47,48]. The ultimate goal for the applied surgical techniques was to eradicate any active endometrial tissues outside its confinements. The hysterolaparoscopic surgical technique plays an integral part by enabling the surgeon to treat or correct anatomical abnormalities found in the uterine cavity which compromise fertility and patient wellbeing such as uterine fibroids (17.8%), Asherman syndrome and synechia (0.8%), polypus (5.4%), uterine septum (6.6%), scar defect (niche; 1.0%), ovarian cyst (0.8%), chronic PID (1.8%) as shown in Table 2. A similar finding was reported by Sreekanth A [26]. During our study, approximately 52.2% of the women had tubal patency restored after peritubal adhesiolysis, 3.5% had partial tubal patency (left or right tube), 4.6% had a total tubal blockage and only 39.7% had normal tubal function.

Most patients had primary infertility (55.2%; 278/504), while secondary infertility accounted for 30.9% (156/504) of cases. Initially, all the women with normal tubal patency had expectant management in the first six months. However, an exception was made for those with blocked tubes, or with other challenges, who were recommended for an assisted reproductive technique (ART) treatment immediately. If pregnancy did not occur during the first six months of follow-up, then the women were asked to proceed to an assisted reproductive technique (ART) treatment for obvious reasons [27,33,40,46]. There were some similarities with other retrospective studies, such as that conducted by Godinjak Z et al., who reported bilateral tubal occlusion in 18 (5%), and unilateral tubal occlusion in 30 (8.33%) patients. Meanwhile, pelvic adhesions were observed in 40 patients (11, 11%) and myomas in 42 (11, 65%), endometrial polyps in 26 (7, 22%) and Syndrome Asherman in 3 patients (0, 83%), etc., by the hysterolaparoscopy procedure. Also, Sreekanth A reported cases of primary infertility in 72% and secondary infertility in 28% of the cases, no tubal pathology in 75% of cases and tubal pathology in 25% of cases, etc. Meanwhile, another study conducted by Nesbitt-Hawes et al. reported that out of the 142/253 (56%) women who attempted to conceive post-surgically, only 104/142 (73%) conceived at the end of the follow-up period with only laparoscopic management, whereas Alborzi A et al. reported 66 spontaneous pregnancies (33.1%), and five (25%) pregnancies through intrauterine insemination [26,27,49,50]. The limitations of these studies are a result of patient desire, withdrawals during the study, heterogeneity of the patients and loss during follow-up. Other limitations include their retrospective nature, population size, and communication challenges. In this study, some patients were referred to our centre solely because of the surgical procedure. However, during the data collection, contact was made to reach out to patients via telephone and emails to compensate for missing information.

## 5. Conclusions

Many studies have shown that endometriosis is sometimes asymptomatic [5,41]. However, problems such as dyspareunia, dysmenorrhea, and chronic pelvic pain can be challenging to patients’ financial, physical and emotional well-being [7,30]. Furthermore, coupling these issues with infertility is very stressful for the patient. Endometriosis is a common medical condition associated with infertility, affecting a woman’s general being. The hysterolaparoscopic surgical management of endometriosis-associated infertility is an effective and relatively safe procedure in infertile women. It is an efficient means to detect and manage other anatomical structural abnormalities often observed in the uterus, as well as the fallopian tubes and the lower pelvis, in a single surgical session. Therefore, we encourage and recommend this procedure as an ideal tool for a fertility work-up, unexplained infertility and endometriosis-related infertility.

## Figures and Tables

**Figure 1 jcm-09-00507-f001:**
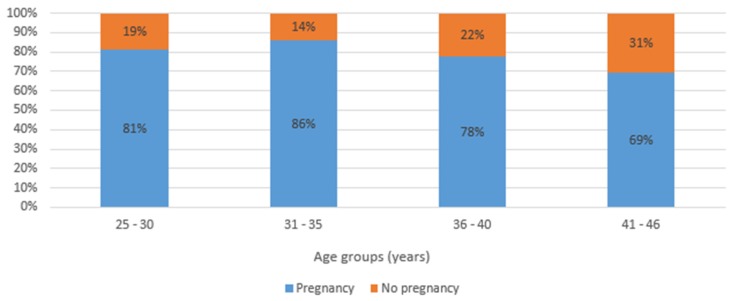
Post-surgical fertility performance of patients with endometriosis-related infertility by age group (*n* = 455).

**Figure 2 jcm-09-00507-f002:**
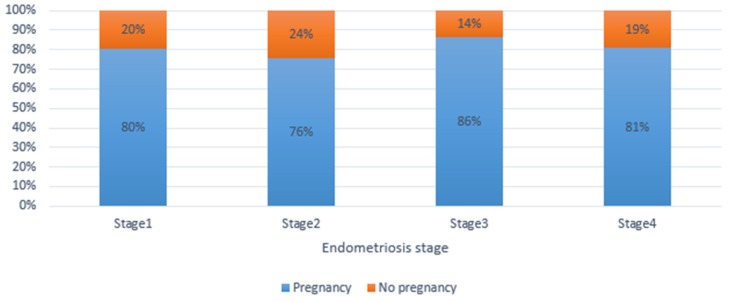
Postsurgical fertility performance by the stages of the endometriosis.

**Table 1 jcm-09-00507-t001:** General characteristics of endometriosis-related infertility patients with a desire to have children after surgery (*n* = 455).

Characteristics	Category	Total (*n* = 455)		No Pregnancy (*n* = 85)		Pregnancy (*n* = 370)		Chi^2^ Test
		*N*	(Col.%)	*N*	(Row%)	*N*	(Row%)	*p*
Age (years)	25-30	91	(20.0%)	17	(18.7%)	74	(81.3%)	0.099
	31-35	179	(39.3%)	25	(14.0%)	154	(86.0%)	
	36-40	159	(35.0%)	35	(22.0%)	124	(78.0%)	
	41-46	26	(5.7%)	8	(30.8%)	18	(69.2%)	
Length of menstrual cycle (days)	≤ 24	93	(20.4%)	18	(19.4%)	75	(80.6%)	0.409
	25-35	338	(74.3%)	65	(19.2%)	273	(80.8%)	
	36+	24	(5.3%)	2	(8.3%)	22	(91.7%)	
Length of menstruation (days)	<4	125	(27.5%)	18	(14.4%)	107	(85.6%)	0.318
	5-6	219	(48.1%)	46	(21.0%)	173	(79.0%)	
	7+	111	(24.4%)	21	(18.9%)	90	(81.1%)	
Stage of endometriosis	1	61	(13.4%)	12	(19.7%)	49	(80.3%)	0.138
	2	132	(29.0%)	32	(24.2%)	100	(75.8%)	
	3	162	(35.6%)	22	(13.6%)	140	(86.4%)	
	4	100	(22.0%)	19	(19.0%)	81	(81.0%)	

**Table 2 jcm-09-00507-t002:** Clinical characteristics and intra-surgical findings of patients with endometriosis-related infertility (*n* = 455).

Characteristics	Category	Total (*n* = 455)	No Pregnancy (*n* = 85)	Pregnancy (*n* = 370)	Chi^2^ Test
		*N* (Col%)	*N* (Row%)	*N* (Row%)	*p*
Causes of infertility	Idiopathic factor	213 (46.8%)	38 (17.8%)	175 (82.2%)	0.001
	Female factor	198 (43.5%)	30 (15.2%)	168 (84.8%)	
	Both parties	44 (9.7%)	17 (38.6%)	27 (61.4%)	
Duration of infertility (years)	1–3 years	339 (74.5%)	61 (18.0%)	278 (82.0%)	0.343
	4–6 years	71 (15.6%)	12 (16.9%)	59 (83.1%)	
	>6 years	45 (9.9%)	12 (26.7%)	33 (73.3%)	
Tubal function	Normal tubal function	239 (52.6%)	47 (19.6%)	193 (80.4%)	0.497
	Normal right tubal function	10 (2.2%)	0 (0.0%)	10 (100.0%)	
	Normal left tubal function	3 (0.7%)	0 (0.0%)	3 (100.0%)	
	Both tubes blocked	20 (4.4%)	3 (15.0%)	17 (85.0%)	
	Normal tube after adhsiolysis	182 (40.1%)	35 (19.2%)	147 (80.8%)	
Type of infertility	Primary infertility	319 (70.1%)	53 (16.6%)	266 (83.4%)	0.083
	Secondary infertility	136 (29.9%)	32 (23.5%)	104 (76.5%)	
Anatomical abnormalities	Adenofibroma	80 (18.2%)	17 (21.3%)	63 (78.8%)	0.469
	Endonmetrial polyp	28 (6.4%)	6 (18.8%)	26 (81.3%)	
	Septum uteri/malformation	39 (8.9%)	6 (15.4%)	33 (84.6%)	
	Ashermann/synechia	5 (1.1%)	1 (20.0%)	4 (80.0%)	
	Chronic PID	9 (2.0%)	1 (11.1%)	8 (88.9%)	
	Ovarium cyst	1 (0.2%)	1 (100.0%)	0 (0.0%)	
	Scar defect (Niche)	6 (1.4%)	0 (0.0%)	6 (100.0%)	
	No	272 (61.8%)	52 (19.6%)	220 (80.9%)	

**Table 3 jcm-09-00507-t003:** Post-surgical fertility performance among 25–35-year-old women with infertility-related endometriosis.

Preoperative ART Treatment	Postoperative ART Treatment	No Pregnancy	Pregnancy	Total
		N (Row%)	N (Row %)	N
No	No	15 (16.3%)	77 (83.7%)	92
	Yes	6 (13.6%)	38 (86.4%)	44
	Total	21 (15.4%)	115 (84.6%)	136
Yes	No	14 (25.9%)	40 (74.1%)	54
	Yes	7 (8.8%)	73 (91.2%)	80
	Total	21 (15.7%)	113 (84.3%)	134
Total		42 (15.6%)	228 (84.4%)	270

**Table 4 jcm-09-00507-t004:** Post-surgical fertility performance among 36–46-year-old women with infertility-related endometriosis.

Preoperative ART Treatment	Postoperative ART Treatment	No Pregnancy	Pregnancy	Total
		N (Row%)	N (Row %)	N
No	No	11 (42.3%)	15 (57.7%)	26
	Yes	3 (16.7%)	15 (83.3%)	18
	Total	14 (31.8%)	30 (68.2%)	44
Yes	No	14 (26.4%)	39 (73.6%)	53
	Yes	15 (17.0%)	73 (83.0%)	88
	Total	29 (20.6%)	112 (79.4%)	141
Total		43 (23.2%)	142 (76.8%)	185

**Table 5 jcm-09-00507-t005:** Type of laparoscopic surgical procedures including CO_2_ laser and fertility performance among women with infertility-related endometriosis (*n* = 455).

Type of Laparoscopic Surgical Procedures Undertaken	No Pregnancy		Pregnancy		Total
	*N*	(Row%)	*N*	(Row%)	
CO_2_ laser not used	29	(16.8%)	144	(83.2%)	173
CO_2_ lazer evaporation technique used	13	(24.1%)	41	(75.9%)	54
Electrocoagulation excision of deep infiltrating endometriosis lesions & adhesiolysis	6	(20.7%)	23	(79.3%)	29
Electrocoagulation excision of superficial peritoneal & deep infiltrating lesion & endometrioma stripping& adhesiolysis	13	(13.7%)	82	(86.3%)	95
Cauterisation of bilateral ovarian endometriosis & Electrocoagulation excision of superficial peritoneal & deep infiltrating lesion & adhesiolysis	2	(16.7%)	10	(83.3%)	12
Endometrioma stripping & Adhesiolysis & Cauterisation of ovary endometriosis	10	(23.8%)	32	(76.2%)	42
Electrocoagulation excision of superficial peritoneal lesion & adhesiolysis	2	(28.6%)	5	(71.4%)	7
Electrocoagulation excision of superficial peritoneal & deep infiltrating lesion & adhesiolysis	12	(27.9%)	31	(72.1%)	43

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
