# Peer review of "The Surgical Benefit of Hysterolaparoscopy in Endometriosis-Related Infertility: A Single Centre Retrospective Study with a Minimum 2-Year Follow-Up"

_jcm, 2020, doi:10.3390/jcm9020507_

Round 1
Reviewer 1 Report
Thank you for the changes to the manuscript. I recommend a final review for minor English language and grammatical errors, including removal of the exclamation marks in the updated results section.
Author Response
The reviewer recommends a final review for minor English language and grammatical errors, including removal of the exclamation marks in the updated results section.
We are very grateful, and we also appreciate Reviewer 1 for her/his for the valuable comments, suggestions and complementation of this work. In response to the areas of concern, we have endeavoured to make the relevant corrections in the affected section of the manuscript; which includes spelling, punctuation, structural adjustment etc. The exclamation marks also removed.
Reviewer 2 Report
My questions have been adequately answered.
I do not agree with conclusions that should be modified:
"Therefore, we convincingly recommend this procedure as an ideal tool for a fertility work-up, unexplained infertility and endometriosis-related infertility, irrespective of invasiveness and costs"
Firstly , the study does not include a cost benefit analysis . Secondly, the hysterolaparoscopy may be recommended in case of pelvic pain. the conclusions should be softened.
Author Response
The review discusses that; Therefore, we convincingly recommend this procedure as an ideal tool for a fertility work-up, unexplained infertility and endometriosis-related infertility, irrespective of invasiveness and costs" Firstly, the study does not include a cost-benefit analysis. Secondly, the hysterolaparoscopy may be recommended in case of pelvic pain. the conclusions should be softened.
We are very grateful, we also thank you dearly Reviewer 2 for your immense support with your comments, and suggestion, in response, we have modified the suggested section of the manuscript. The hysterolaparoscopy surgical management of endometriosis-associated infertility is an effective and relatively safe procedure in infertile women. It is an efficient means to detect and manage other anatomical structural abnormalities often observed in the uterus, as well as fallopian tubes and the lower pelvis, in a single surgical session. Therefore, we encourage and recommend this procedure as an ideal tool for a fertility work-up, unexplained infertility and endometriosis-related infertility.
This manuscript is a resubmission of an earlier submission. The following is a list of the peer review reports and author responses from that submission.
Round 1
Reviewer 1 Report
This is a retrospective study looking at pregnancy rates after hysteroscopic and laparoscopic surgery for all stages of endometriosis at a single centre. This study adds to the body of data that already exists in this area, however I have some concerns about the way the data is presented and the conclusions drawn as a result. See my detailed comments below:
Abstract:
Line 20 – “Patients aged ≤ 35 years had a better fertility outlook post-surgery (p = 0.039)” This statement is meaningless without more information so needs to be backed up with numbers even in the abstract or removed from the abstract.
Line 28 – “These factors positively affect quality of life and social wellbeing” This statement cannot be concluded from the results of your study.
Introduction:
There needs to be more on the previous research performed addressing fertility outcomes for laparoscopic excision of endometriosis here. You address some in the discussion, but I think it is important as part of the background to the study.
Materials and Methods:
Line 88: “et cetera”. Either state the actual investigations or remove this part.
Results:
Line 128: “(≤ 35 and > 35 years)” this is not applicable as in your graph you divide the patients into four age categories
Figure 1: Percentages should be recorded on the figure as well as frequencies
Table 1: Are you trying to compare demographic details between your groups of “became pregnant” and “did not conceive”? In that case, the table is not correct in the reporting of the percentages. Should be more like this:
|
|
No pregnancy |
Pregnancy |
This column should be a significance factor when comparing the two… |
|
≤35 |
34/85 (40%) |
199/370 (54%) |
|
|
≥36 |
51/85 (60%) |
171 (46%) |
|
Table 2 also has this issue and needs denominators in the fields
Figure 3 is confusing: You state this is relative pregnancy by age group and that it is statistically significant, but I am not sure what you are trying to demonstrate. Obviously women with no pregnancy will have a lower frequency than the group that achieved a pregnancy. What exactly is significant?
Your results do not indicate how many women had IVF or whether they tried to conceive naturally first and then progressed to IVF. This is an important distinction as the first 6-12 months have been demonstrated to be the highest chance time to conception after resection of endometriosis.
Line 152 – please include the raw numbers, not just percentages
Line 155 – you cannot state that it was surgical expertise as you cannot quantify your own expertise. You can state that this:
“finding indicates that the fertility performance was not greatly influenced by the age of women, but rather by surgical removal of endometriosis”
Incidentally, you do not comment on completeness of resection versus incomplete resection. Do you have these data?
Line 159 – “There was significant difference in the pregnancy outcome in the ≤ 35-year-old group with regards to spontaneous pregnancy (74.1%) compared to ART-assisted pregnancy (91.3%) post surgery (P = 0.007; OR = 3.7; 95% CI = 1.4-9.8)”
This does not make sense to me – in the ≤35 year old age group 74% had a spontaneous pregnancy and 91% had an ART pregnancy? These numbers do not add up. If you mean that of those who tried to conceive naturally 74% achieved a pregnancy, compared to those who tried ART where 91% achieved a pregnancy, this needs to be stated more clearly. And with numerators and denominators/raw numbers not just percentages.
Discussion:
Line 173: “This retrospective study highlights that early expert minimal surgical treatment of endometriosis-related infertility is the most feasible treatment option” This statement cannot be concluded from your study. What do you term early? How does your study show that early treatment is helpful? I also recommend removing ‘expert’ as it is impossible to state this without talking about completeness of resection etc.
You should not be including new data in the discussion that is not previously included in the results.
Line 228: “and myomectomy in cases of adenomyosis” – perhaps you mean adenomyomectomy? A myomectomy would be performed in cases of concurrent leiomyomata.
You should discuss the limitations of your study – retrospective nature, incomplete follow up…
Please also discuss the high success rates of pregnancy achieved eg 91% of patients in the <35 ART group. This seems higher than expected
Conclusions:
Line 255: “time efficient and safe procedure” – you cannot state this without backing it up with data
Author Response
Reviewers 1
Those who were unable to achieve spontaneous pregnancy after six months follow up post-surgery were admitted into the ART treatment program, and except those with complete tubal blockage who commenced ART treatment immediately.
Statement of effectivity and safety of the procedure is borne out of the fact that both surgeries are at the same sitting. Thereby, saving patients from undergoing many rigorous investigations like (HSG, Hyscoy, diagnostic hysteroscopy). After which they may require laparoscopy evaluation before possible ART treatment. With this procedure, patients are evaluated once, lesser exposure to anaesthesia, and multiple financial implications avoided.
We appreciate the contribution made to this article and all the advice. We also hope, we are able to provide more information, that may be missing initially.
Reviewer 2 Report
In this study, Ekine et al. investigated the surgical benefit oh hysterolaparoscopy in endometriosis-related infertility. It is a retrospective single centre study. The manuscript is well written and the study is well organized and comprehensively described . However, I have some questions:
Table 1. In the last row the authors have shown the mode of pregnancy. I can’t understand why in the column of without pregnancy on the left they are included patients with pregnancy.
Table 2. Women included was all affected by endometriosis : why were 213 patients classified as idiopathic factor? Which kind of patients did “female infertility” include?
Moreover , the selection of the patients is not totally clear: male factor probably should be excluded
Discussion: The authors declare that patients underwent different type of surgery: stripping, laser etc.. from evidence in literature different surgical techniques may induce different reulsts. The authors should divide population on the basis of the type of surgery.
They routinely did hysteroscopy and they had corrected anatomical defects as fibroids, polyps, adhesyons and septa: this is a bias. The removal of anatomical defects improve pregnancy per se independently from endometriosis, they should exclude them from analysis.
Author Response
Reviewers 2
We tried to make some modifications in those areas you commented upon.
Included also is the type of surgical procedures as request, which was initially intentionally omitted to reduce the length of the article. Finally, your comment was very appreciative. In addition, we also modified the figures and added additional tables for clarity.
We appreciate the contribution made to this article and all the advice given. We also hope, we were able to provide more information, which may be missing initially.
Round 2
Reviewer 1 Report
Although there have been revisions made to the manuscript, I still do not feel they are sufficient to warrant publication in its current form.
Further suggestions:
In general your revisions appear a bit haphazard and rushed. There are multiple spelling mistakes and formatting errors. There are areas where you have repeated yourself in the reporting of results.
Line 18 – The statement of 94% of pregnancies resulting in live birth is much higher than expected in the normal population. Can you explain this in your discussion? Maybe at line 212
Line 23 – Change “better fertility outlook” to “higher chance of conception”
Figure 1 and 2 – I think this would be more meaningful if you reported the percentages, not the actual patient numbers eg 25-30 group 19% not pregnant (rather than 17 women) and 81% pregnant (rather than 74 women).
Tables – your tables are all difficult to interpret. Please remove the columns at the right which illustrate the totals. You can put the data in as a denominator within the other columns instead. Please be clear on what you are trying to report. Are you trying to compare women with pregnancy vs no pregnancy? In this case, please report your p values/confidence intervals in the final column to show how they differ if at all. Your table legends should not include results/data. Table 5 includes too many subsets of surgery to be meaningful. Consider grouping these further to reduce the number of rows
Line 183-185 – Please report the actual p value rather than P > 0.05
Line 195 – “At the end of the follow up period…” It appears that you have not completed this sentence.
Line 246-247 – “in addition to other co-joining factors of individuals of the study groups” – this sentence does not make sense
Line 278-279 – You cannot just state there were similarities of these other papers – what did they show?
Line 279 – “Erin M et al” should read “Nesbitt-Hawes et al” – the reference section for this paper also needs to be corrected
You haven’t included up to date references with none since 2016. As such you have not discussed the endometriosis fertility index and how it might relate to your population given you did not find a difference in fertility outcomes for various stages of endometriosis.
Reviewer 2 Report
The authors did not answer or modify the text as asked. The revision was not satisfying and the results are not discussed as requested.